# Tumor Tissue-Specific Biomarkers of Colorectal Cancer by Anatomic Location and Stage

**DOI:** 10.3390/metabo10060257

**Published:** 2020-06-19

**Authors:** Yuping Cai, Nicholas J. W. Rattray, Qian Zhang, Varvara Mironova, Alvaro Santos-Neto, Engjel Muca, Ana K. Rosen Vollmar, Kuo-Shun Hsu, Zahra Rattray, Justin R. Cross, Yawei Zhang, Philip B. Paty, Sajid A. Khan, Caroline H. Johnson

**Affiliations:** 1Department of Environmental Health Sciences, Yale School of Public Health, Yale University, New Haven, CT 06520, USA; ping.cai@yale.edu (Y.C.); nicholas.rattray@strath.ac.uk (N.J.W.R.); qian.zhang.qz235@yale.edu (Q.Z.); varvara.mironova@yale.edu (V.M.); alvarojsn@iqsc.usp.br (A.S.-N.); ana.rosenvollmar@yale.edu (A.K.R.V.); yawei.zhang@yale.edu (Y.Z.); 2Strathclyde Institute of Pharmacy and Biomedical Sciences, University of Strathclyde, Glasgow G4 0RE, UK; zahra.rattray@strath.ac.uk; 3São Carlos Institute of Chemistry, University of São Paulo, São Carlos 13566-590, SP, Brazil; 4Department of Surgery, Memorial Sloan Kettering Cancer Center, New York, NY 10065, USA; mucae@mskcc.org (E.M.); hsuk1@mskcc.org (K.-S.H.); patyp@mskcc.org (P.B.P.); 5Cancer Biology and Genetics Program, Memorial Sloan-Kettering Cancer Center, New York, NY 10065, USA; crossj@mskcc.org; 6Department of Surgery, Yale University School of Medicine, New Haven, CT 06520, USA; 7Department of Surgery, Division of Surgical Oncology, Yale University School of Medicine, New Haven, CT 06520, USA

**Keywords:** metabolomics, colorectal cancer (CRC), metabolite biomarkers

## Abstract

The progress in the discovery and validation of metabolite biomarkers for the detection of colorectal cancer (CRC) has been hampered by the lack of reproducibility between study cohorts. The majority of discovery-phase biomarker studies have used patient blood samples to identify disease-related metabolites, but this pre-validation phase is confounded by non-specific disease influences on the metabolome. We therefore propose that metabolite biomarker discovery would have greater success and higher reproducibility for CRC if the discovery phase was conducted in tumor tissues, to find metabolites that have higher specificity to the metabolic consequences of the disease, that are then validated in blood samples. This would thereby eliminate any non-tumor and/or body response effects to the disease. In this study, we performed comprehensive untargeted metabolomics analyses on normal (adjacent) colon and tumor tissues from CRC patients, revealing tumor tissue-specific biomarkers (*n* = 39/group). We identified 28 highly discriminatory tumor tissue metabolite biomarkers of CRC by orthogonal partial least-squares discriminant analysis (OPLS-DA) and univariate analyses (VIP > 1.5, *p* < 0.05). A stepwise selection procedure was used to identify nine metabolites that were the most predictive of CRC with areas under the curve (AUCs) of >0.96, using various models. We further identified five biomarkers that were specific to the anatomic location of tumors in the colon (*n* = 236). The combination of these five metabolites (S-adenosyl-L-homocysteine, formylmethionine, fucose 1-phosphate, lactate, and phenylalanine) demonstrated high differentiative capability for left- and right-sided colon cancers at stage I by internal cross-validation (AUC = 0.804, 95% confidence interval, CI 0.670–0.940). This study thus revealed nine discriminatory biomarkers of CRC that are now poised for external validation in a future independent cohort of samples. We also discovered a discrete metabolic signature to determine the anatomic location of the tumor at the earliest stage, thus potentially providing clinicians a means to identify individuals that could be triaged for additional screening regimens.

## 1. Introduction

Colorectal cancer (CRC) is the second most common cause for cancer deaths worldwide, with an annual mortality of 862,000 [1]. The survival rate for CRC patients is dependent on American Joint Committee on Cancer (AJCC) stage, where longer survival is seen in those with earlier stage tumors [2]. Therefore, it is of the utmost importance to identify lesions at an early stage so that patients can receive appropriate treatment expeditiously. In the U.S., screening is currently recommended for those aged >50 years, however, the American Cancer Society has recommended lowering this age to 45, due to the recent increase in advanced-stage early-onset CRC cases before the age of 50 [3]. Current screening methods utilized globally include: fecal occult blood tests (FOBTs) (which include fecal immunochemical test (FIT) or guaiac FOBT), stool DNA testing (FIT-DNA), sigmoidoscopy, or standard colonoscopy. Patients that follow the recommended screening guidelines have a lowered risk of death from CRC, however, there are barriers to screening, and <50% of individuals are up-to-date with the recommendations. Some of these self-reported barriers include fear or worry of the procedure, financial hardship, and logistical challenges such as transportation to medical providers [4]. Therefore, the utilization of alternative, less-invasive screening methods such as blood-based biomarker tests would alleviate some of these barriers and identify those that may require further screening or treatment.

A number of studies have attempted to use untargeted metabolomics to identify serum, plasma and urinary metabolite biomarkers for CRC, with the ultimate aim being to develop a blood or urine-based diagnostic test for screening [5,6]. However, these studies have not resulted in identifying a panel of biomarkers that can be reproducibly validated between laboratories or sample cohorts. Levels of some of the biomarkers have even been contradictory between studies; for example, alanine was demonstrated to be increased in urine and serum in two studies [7,8], but it was reported to be decreased in serum in another study [9]. This problem of reproducibility for biomarker discovery is not only contained within CRC studies, but throughout the field of biomarker discovery using metabolomics. In fact, the only clinical application of a blood-based clinical biomarker identified by metabolomics is trimethylamine-N-oxide (TMAO). TMAO is a metabolite co-metabolized from gut bacteria, and it can be measured in blood or urine. It has been linked to the development of atherosclerosis and heart disease complications [10,11]. The lack of clinical applicability stems from lack of study reproducibility. This is because of the difficulty in controlling physiological confounding variables, such as body mass index (BMI), age, gender, and also differences in sample collection and preparation, analytical instrumentation, and data analysis, which have been carried out in biofluids. However, a major problem often discussed, but not addressed, is confounding non-tumor related metabolites that are derived from the diet (fasting vs. fed, type of food), or are related to the effects of circadian rhythm, exercise, and possible co-morbidities. Also, a large dilution effect can occur to the tumor-originating metabolites once they enter the circulatory system, due to the larger pool of non-tumor originating metabolites at higher concentrations in the blood. It may therefore be difficult to find predictive metabolites from blood without ascertaining their levels in the tumors first. Therefore, looking at the localized tumor-specific metabolites first, then extrapolating to measure the same metabolites in blood or urine from CRC patients, would result in a higher likelihood of finding specific and functional clinical biomarkers. 

Limited studies have investigated the anatomic differences in the colon that could affect the metabolism of the primary tumor [12,13]. Within the colon, tumors can form in different anatomic regions. Lesions that develop on the proximal side of the colon (cecum, ascending colon and hepatic flexure) are defined as right-sided colon cancers (RCCs), and those that develop on the distal side of the colon (splenic flexure, descending, sigmoid and rectosigmoid) are left-sided colon cancers (LCCs) [14]. Mounting evidence from epidemiologic studies have shown that clinical outcomes differ for patients with RCCs compared with LCCs [15,16]. A recent meta-analysis comprising 66 studies and more than 1.4 million patients demonstrated that patients with RCC have worse overall prognosis than those with LCC, even after stage matching [6]. Moreover, having a tumor originating in the left side of the colon was associated with a 19% reduced risk of death from the disease [17]. Therefore, identifying anatomic location-specific metabolite biomarkers could help indicate whether an individual may have the disease and direct colonoscopy screening protocols to diagnose the patient. From a clinical perspective, RCC is difficult to diagnose early, as patients do not present with symptoms such as rectal bleeding or anemia until later stages, and the survival rate decreases as stage of diagnosis increases. Therefore, it reinforces the need to find molecular biomarkers for early diagnosis. Previous in vitro metabolomics studies have observed differences in the cholesterol biosynthesis pathway when comparing cell lines originating from patients with RCC (HCT 116) and LCC (DLD-1), treated with calcitriol [18]. The results suggested separate mechanisms of tumorigenesis for RCC and LCC. In addition, plasma metabolomics analysis from RCC and LCC in CRC patients recently reported six potential biomarkers (anserine, trimethylamine N-oxide, arginine, gamma-glutamyl-gamma-aminobutyraldehyde, indoxyl sulfate, and pyridoxal 5′-phosphate), however these, again, suffer from the issue of plasma biomarker discovery and subsequent validation [19].

In this study, a retrospective metabolomics data analysis was conducted on colon tissues (healthy tissue and tumor) from CRC patients, to identify the tissue-specific biomarkers of CRC. An additional focus was to examine the influence of anatomic location of the primary tumors. Initially, an untargeted metabolomics data analysis was performed on normal colon tissues (*n* = 39) and primary colon tumor tissues (*n* = 39, stage I–III), collected from CRC patients to discover tissue-specific metabolic signatures. Then, comparative analyses were applied on a large cohort of RCC and LCC patients (*n* = 236), to further find potential anatomic location-specific metabolic biomarkers.

## 2. Results

### 2.1. Metabolic Differences between Colon Cancer and Normal Controls

Untargeted metabolomics were initially performed on normal adjacent colon (*n* = 39) and primary colon tumor tissues (*n* = 39, stage I–III), collected from CRC patients. The rationale for this analysis was to identify metabolites that were dysregulated with respect to tumor tissues. Detailed information on the clinical cohort is shown in Appendix A. The data generated was subjected to principal components analysis (PCA) and demonstrated a high level of analytical reproducibility, as revealed by the tight clustering of QC samples (Appendix A). A supervised multivariate method using orthogonal partial least-squares discriminant analysis (OPLS-DA) was subsequently applied to reveal global differences in the metabolome between normal and tumor tissues. The OPLS-DA model revealed an excellent separation of samples that were classified as either normal or tumor (R^2^Y = 0.96, Q^2^ = 0.77) (Figure 1a). Permutation tests were carried out by comparing the goodness of fit (R^2^ and Q^2^) of the OPLS-DA models with the goodness of fit of 200 Y-permutated models and demonstrated the validity of our constructed supervised model (Q^2^ intercept −0.08) (Appendix A). Univariate analyses by Wilcoxon Mann–Whitney U test with false discovery rate (FDR) adjustment was also applied to find significantly altered features between normal colon and colon tumor tissues. Twenty-eight metabolites were identified that met with three criteria, a variable importance projection (VIP) >1.5 from the OPLS-DA analysis, with a FDR-corrected *p*-value < 0.05, and fold change > 1.2 or < 0.8 from univariate analysis, and are considered as potential metabolic biomarkers for CRC (Figure 1b,c). Of note, large metabolic differences were shown to exist between normal and tumor tissues; there, a stringent threshold of VIP >1.5 was applied to identify the most discriminatory metabolites, to use as potential clinical biomarkers. The metabolomics standards initiative (MSI) identification level and detailed information on the 28 tissue metabolites are shown in Appendix A. 

To further evaluate the predictive performance of the 28 potential biomarkers, three different models were used with a seven-fold cross validation strategy; partial least-squares (PLS), random forest (RF), and support vector machine (SVM) models. These models were constructed, and the area under the curve (AUC) from the receiver operating characteristic (ROC) curves were computed. Notably, all three models exhibited excellent discriminative performance for CRC based on the 28 biomarkers (Figure 1d). The AUC values were 0.96 (95% confidence interval, CI 0.91–1.00), 0.96 (95% confidence interval, CI 0.92–1.00), and 0.95 (95% confidence interval, CI 0.90–1.00), for PLS, RF, and SVM, respectively. Notably, creatinine [7,8], taurine [20], lactate [8,21], glutamate [19], CDP-choline [20], and glycerol-3-phosphate [22] have been previously reported as correlated to colon cancer from the analysis of urine, serum, or plasma samples from colon cancer patients.

Given that a biomarker panel of 28 metabolites would be challenging to use for clinical application, we carried out stepwise selection on the 28 potential metabolic biomarkers to identify a discrete number that are predictive. The 28 metabolites were sorted according to their VIP values, then, the backward stepwise regression procedure was performed by removing the metabolite with the smallest VIP in each step. In each step, a PLS prediction model was constructed using the remaining metabolites. The bootstrap method was used to sample the dataset [23]. In brief, 63% randomly selected patients from the dataset were selected as discovery data to build the prediction model, and the remaining 37% patients were used as validation data. This random sampling with model construction and validation procedure was repeated 1000 times, and the AUCs were calculated and recorded (Appendix A). The combination of the top nine metabolites ranked by VIP values (Appendix A; taurine, glutamate, CDP-choline, fructose 6-phosphate, hypoxanthine, phenylalanine, phosphoethanolamine, creatinine, and GDP-glucose) was able to construct predictive models and had an excellent AUC value of 0.98 (95% confidence interval, CI 0.95–1.00), 0.97 (95% confidence interval, CI 0.93–1.00), and 0.96 (95% confidence interval, CI 0.91–1.00) for RF, PLS, and SVM, respectively (Appendix A). Of note, the top nine metabolites resulted in the highest AUC value, and increasing the number of biomarkers did not improve the AUC value. To validate the nine CRC metabolic markers, we further assessed their predictive performance in another set of CRC tumor samples, using PLS, RF, and SVM models (Appendix A). The results demonstrated that all three models exhibited excellent discriminative performance for CRC based on the nine biomarkers (Appendix A). The AUC values were 0.86 (95% confidence interval, CI 0.76–0.96), 0.93 (95% confidence interval, CI 0.88–0.99), and 0.91 (95% confidence interval, CI 0.84–0.98), for PLS, RF, and SVM, respectively.

### 2.2. Distinct Metabolomic Signatures Associated with Anatomic Location

To investigate the influence of tumor location on the metabolome, we analyzed a larger number of colon tumor tissues from CRC patients (*n* = 197), which included the 39 tumor samples analyzed in the previous section. The detailed information on this clinical cohort is shown in Appendix A. The PCA conducted on the patient samples and QCs showed tightly clustered QC samples (Appendix A). OPLS-DA models were used to examine tumor stage-specific differences between RCCs from LCCs. In addition, the stage II model was constructed by taking tumor size into account; since differences in tumor size were observed between RCC and LCC, larger tumors may exhibit areas of hypoxia which could affect metabolism (Appendix A). As the OPLS-DA plots demonstrate in Figure 2, distinct metabolic profiles were observed between RCC and LCC patients when stratified by stage. Meanwhile, permutation tests demonstrated that all three OPLS-DA models were well fitted, as the permutated R^2^ and Q^2^ values were both smaller than the experimentally-derived values, meaning that the models are significant and predictive (Appendix A). Although separation between RCCs and LCCs was also seen visually by integrating tumors across stages I–III, the statistical model was not validated by the permutation test in which the permutated R^2^ value was larger when experimentally calculated (Appendix A). The Q^2^ value in the PLS model is often used to assess predictive ability and is calculated using a blindfolded procedure, and a model with Q^2^ > 0 is regarded as having predictive relevance [24]. Among the three OPLS-DA models, the resulting model from stage I tumors had the best predictive ability, with a Q^2^ value of 0.45. The stage II model had a lower Q^2^ value of 0.23, and stage III has a Q^2^ value of −120.27; the latter therefore showing poor predictive ability. The R^2^Y values, which are the fraction of the variation of Y variables explained by the models, are 0.96, 0.87, and 0.91 for stage I, and stage II, and stage III, respectively. The results indicate that the metabolomics profiles for stage I patients can discriminate between those with early-stage RCC and LCC, indicating potential biomarkers for early stage CRC by lesion location.

### 2.3. Potential Metabolic Biomarkers for Early-Stage LCC and RCC

To identify metabolic biomarkers for early-stage (stage I) LCC and RCC, we next incorporated normal controls into the biomarker discovery procedure. This was to ensure that the markers were not only markers of tissue location, but were also markers of cancer in these different regions. The criteria for the selection of potential metabolic biomarkers were set as follows: (1) VIP value (OPLS-DA) > 1.0; (2) *p* value (LCCs vs. RCCs) < 0.05; (3) *p* value (normal colon tissues from left-side vs. normal colon tissues from right-side) >0.05; (4) *p* value (normal colon tissues from left-side vs. LCCs) < 0.05 or *p* value (normal colon tissues from right-side vs. RCCs) < 0.05. The VIP criteria (>1.0) was different from above one (>1.5), ascribing to a large difference between normal colon and tumor, and relative smaller difference between RCC and LCC. A total of five metabolites met with above-mentioned criteria and were selected as potential metabolic biomarkers for LCCs and RCCs. These metabolites are S-adenosyl-L-homocysteine (SAH), formylmethionine, fucose 1-phosphate, lactate, and phenylalanine. The detailed information is displayed in Table 1.

As shown in Figure 3a, three metabolites (SAH, formylmethionine, and fucose 1-phosphate) were consistently upregulated in tumor tissues compared to normal tissues from the same side of the colon, and could be considered as potential biomarker candidates of stage I CRC for either LCC or RCC. However, two additional metabolites were only increased in LCCs, lactate and phenylalanine. Of note, no difference (*p* > 0.05) was observed between left-sided normal colon and right-sided normal colon for the five discovered potential biomarkers (Table 1). We further developed a random forest model with the five potential biomarkers, to investigate the predictive performance in determining whether a tumor is stage I LCC or RCC. The combination of these five metabolites demonstrated a good differentiative capability (Figure 3b). The AUC value of the five metabolites combined is 0.804 (95% confidence interval, CI 0.670−0.940), with a sensitivity of 86.4% and specificity of 68.0% for stage I LCC. It is worthy to note that these five metabolites are also general potential biomarkers for colon cancer, regardless of stage or anatomic location, as we demonstrated in our comparison of normal colon to colon tumor tissues. However, as can be seen here, some caution may be needed when delving into specific subgroups such as stage I RCC, as the markers may have lower specificity. Therefore, we performed an additional examination. A further stratification by tumor stages (stage I, stage II, and stage III) and anatomic location were conducted on the five anatomic-specific metabolic biomarkers. Among them, SAH, formylmethionine, and fucose 1-phosphate were up-regulated in patients with LCC and RCC across all stages, (*p* < 0.05) (Figure 4). Lactate was increased only in LCCs at stage I, and it was not changed in LCCs at stages II and III, compared with normal control tissues from the left side of the colon. However, lactate was up-regulated in stage III RCCs compared to normal right-sided colon tissues. A similar difference was seen for the essential aromatic amino acid phenylalanine; this metabolite was only increased in LCCs at stage I, and not changed in RCCs at any stages compared to normal tissues.

## 3. Discussion

Altered metabolism is a significant feature of CRC, and dysregulated metabolites are associated with colon cancer development, prognosis, and recurrence [9,25,26,27]. A number of studies have used untargeted metabolomics to identify metabolite biomarkers for CRC from various accessible biofluids collected from patients, such as plasma, serum, and urine [5,6]. The most commonly identified metabolite biomarkers for CRC from these studies were amino acids (alanine, glutamine, glutamate etc.), carbohydrates (lactate, pyruvate etc.), and fatty acids (butyrate, propionate etc.) Generally, a panel of metabolite biomarkers demonstrated better diagnostic performance than a single biomarker. However, these studies have been hampered by lack of reproducibility, and in some cases, results are contradictory. In our previous work, an untargeted metabolomics analysis was performed on normal colon and primary colon tumor tissues collected from a large cohort of CRC patients, to identify sex-associated differences in colon cancer metabolism [13]. In this study, we carried out a retrospective metabolomics data analysis to identify tissue-specific biomarkers of CRC from this cohort. Our first aim was to identify highly specific CRC biomarkers through the examination of normal colon and primary colon tumor tissues. In total, we identified 28 metabolites which demonstrated excellent discriminative performance by machine learning based PLS, SVM and RF models. We then used a backward stepwise regression method to identify the nine most discriminatory metabolites that could be used as a biomarker panel for testing in the blood or urine from CRC patients.

Of note, four out of nine metabolites we identified are perfectly consistent with the results from various metabolomics studies using biofluid or tissue samples. Taurine and glutamate, which are with the highest VIP values in the OPLS-DA model, are consistently upregulated in different CRC study cohorts with tumor tissue derived from tissue, plasma, and serum [20,28,29,30]. Additionally, our study identified CDP-choline, a dispensable intermediate for the synthesis of structural phospholipids of cell membranes, and had a VIP of 2.22. This metabolite is also increased in plasma samples from other CRC studies [20]. The down-regulation of creatinine which we identified, is in accordance with other studies using urine, plasma, and serum samples [8,20,31]. Given that these potential biomarkers were readily measured and validated in biofluid samples, it is highly plausible that the metabolites our study identified is of potential value to develop clinically applicable markers for CRC diagnosis with further validation in different bio-fluid sample cohorts. In addition, further studies that include controls from other intestinal diseases such as colitis, Crohn’s disease, diverticulitis, and also from other types of tumor, are required to demonstrate the specificity of metabolite biomarkers for colorectal tumorigenesis. Moreover, it is highly recommended to investigate the specificity of metabolite biomarkers in a newly collected biobank, since one limitation of the current study might be that metabolite degradation could happen during long-term storage.

Since anatomic tumor location is associated with symptoms, screening efficacy, and patient survival, we carried out metabolomics analysis, stratified by AJCC stage, on a large cohort of patients with either RCC or LCC, to identify metabolic biomarkers that differentiate tumors based on location. We identified five metabolites (SAH, formylmethionine, fucose 1-phosphate, lactate, and phenylalanine) which are specific for stage I LCC. This combination of these five biomarkers demonstrated a good predictive power for differentiating between stage I RCCs and LCCs with AUC value 0.804 (95% confidence interval, CI 0.670–0.940). Importantly, these metabolites were specific for cancer tissue, and were not different when comparing healthy right-sided colon and left-sided colon. 

Patients with RCC have worse prognosis than those with LCC, and RCC is less likely to be diagnosed at an early stage. Thus, the clinical applicability of the panel of five biomarkers identified in our study may prove to be useful for medical providers. To explore the clinical application of the five metabolites, their abundance level and limits of detection in blood or urine samples need to be further examined, because of the dilution effect in biofluids. Furthermore, validation in independent cohorts with biofluid samples, comparing patients with RCC and LCC at early stages, is required. Finally, future survival analyses could be valuable, to reveal the association between the five metabolite biomarkers identified in our study and the prognosis of patients with RCC or LCC.

In conclusion, our data has revealed a biomarker panel of nine metabolites specific for CRC. We also identified tumor location-specific biomarkers for stage I CRC with potential clinical implications. Future studies examining the clinical utility of these biomarkers in biofluids for the non-invasive screening of early stage colon cancer are underway.

## 4. Materials and Methods

### 4.1. Chemicals

Ammonium acetate and formic acid were purchased from Fisher Scientific (Morris Plains, NJ, USA). Ammonium hydroxide was purchased from Honeywell (Muskegon, MI, USA). Water (H_2_O), methanol (MeOH) and acetonitrile (ACN) were LC-MS grade and were purchased from Fisher Scientific (Morris Plains, NJ, USA).

### 4.2. Sample Collection

Colon tumor and normal colon tissue (away from the tumor at the resection margin) were acquired from surgical colectomy specimens and prospectively collected on 736 stage I–IV CRC patients in the period 1991–2001 at Memorial Sloan-Kettering Cancer Center (MSKCC, New York, NY, USA). Clinical data were recorded and updated retrospectively. Each sample was snap frozen in liquid nitrogen and immediately stored at −80 °C. Pre-operative intravenous antibiotics (cefazolin/metronidazole, clindamycin/gentamicin or ciprofloxacin/metronidazole) were administered within 60 min prior to resection. All patients received a standard mechanical bowel preparation (polyethylene glycol (PEG) solution) 24 h before scheduled surgery. For this study, samples were selected from patients that were ≥55 years old, to reduce the confounding effects of estrogen signaling on metabolism before menopause. All normal colon tissues were selected from stage I–IV CRC patients (*n* = 39), and tumor tissue samples were selected from RCCs and LCCs stage I–III (*n* = 197). Stage IV tumor samples were not included, as their metabolism may be affected by the presence of metastases in the liver or other site, therefore we cannot rule out this confounder. Normal adjacent tissues were from stage I–IV patients. Normal colon tissue samples were taken from stage IV patients, as their metabolic profiles were not significantly different from the normal ones taken from stage I–III (Appendix A), and they were markedly different from the tumor samples. The Yale University Institutional Review Board (IRB) determined that the study conducted in this publication was not considered to be Human Subjects Research and did not require an IRB review (IRB/HSC# 1612018746).

### 4.3. Tissue Metabolite Extraction

First, 50 ± 1 mg of each tissue was homogenized using 500 μL of UPLC-grade H_2_O. A Cryolys Evolution homogenizer (Bertin Corporation, Rockville, MD, USA) was used with 2 mL lysing tube (Bertin Corporation) and 1.4 mm ceramic zirconium oxide beads (Bertin Corporation) to homogenize the tissues. Each sample was processed six times for 20 s, at 6000 rpm with 5 s intervals. Dry ice was used to keep the temperature <10 °C during homogenization. From the homogenized solution, 100 µL was taken and added to 1.5 mL polypropylene microcentrifuge tubes for subsequent metabolite extraction. A volume of 400 μL ice cold MeOH:ACN (1:1, *v*/*v*) was added to each sample as the extraction solvent. The samples were vortexed for 30 s, and sonicated for 10 min. To precipitate proteins, the samples were incubated for 2 h at −20 °C, followed by centrifugation at 13,000 rpm (15,000× *g*) and 4 °C for 15 min. The resulting supernatant was removed and evaporated to dryness for 12 h using a vacuum concentrator (Thermo Fisher Scientific, Waltham, MA, USA). The dry extracts were then reconstituted in 100 µL of ACN:H_2_O (1:1, *v*/*v*), sonicated for 10 min, and centrifuged at 13,000 rpm (15,000× *g*) and 4 °C for 15 min, to remove insoluble debris. The supernatant was transferred to UPLC autosampler vials (Thermo Scientific, Waltham, MA, USA). A pooled quality control (QC) sample was prepared by mixing 5 μL of extracted solution from each sample into a UPLC autosampler vial. All the vials were capped and stored at −80 °C prior to UPLC-MS analysis.

### 4.4. UPLC-MS Analysis

Both hydrophilic interaction chromatography mass spectrometry (HILIC-MS) and reverse phase liquid chromatography mass spectrometry (RPLC-MS) approaches were used for a comprehensive analysis of the tissue metabolome. A UPLC system (H-Class ACQUITY, Waters Corporation, Milford, MA, USA), coupled to a quadrupole time-of flight (QTOF) mass spectrometer (Xevo G2-XS QTOF, Waters Corporation, Milford, MA, USA), was used for MS data acquisition. A Waters ACQUITY UPLC BEH Amide column (particle size, 1.7 μm; 100 mm (length) × 2.1 mm (i.d.)) and Waters ACQUITY UPLC BEH C18 column (particle size, 1.7 μm; 50 mm (length) × 2.1 mm (i.d.)) were used for the UPLC-based separation of metabolites. The column temperature was kept at 25 °C for HILIC-MS analysis and 30 °C for RPLC-MS analysis. The solvent flow rate was 0.5 mL/min, and the sample injection volume was 1 μL. For HILIC-MS analysis, mobile phase A was 25 mM NH_4_OH and 25 mM NH_4_OAc in water, while the mobile phase B was can, for both electrospray ionization (ESI), positive and negative mode, respectively. The linear gradient was set as follows: 0~0.5 min: 95% B; 0.5~7 min: 95% B to 65% B; 7~8 min: 65% B to 40% B; 8~9 min: 40% B; 9~9.1 min: 40% B to 95% B; 9.1~12 min: 95% B. For RPLC-MS analysis, the mobile phases A was 0.1% formic acid in H_2_O, while the mobile phases B was 0.1% formic acid in ACN, respectively, for both ESI+ and ESI−. The linear gradient was set as follows: 0~1 min: 1% B; 1~8 min: 1% B to 100% B; 8~10 min: 100% B; 10~10.1 min: 100% B to 1% B; 10.1~12 min: 1% B. Pooled samples were analyzed every eight injections during the UPLC-MS analysis to monitor the stability of the data acquisition, and used for subsequent data normalization.

QTOF scan data (300 ms/scan; mass scan range 50–1000 Da) were initially acquired for each biological sample for metabolite quantification. Then, both DDA (data-dependent acquisition) data (QTOF scan time: 50 ms/scan, MSMS scan time 50 ms/scan, collision energy 20 eV, top 5 most intense ions were selected for fragmentation, exclude former target ions (4 s after 2 occurrences)) and MS^E^ data (low energy scan: 200 ms/scan, collision energy 6 eV; high energy scan: 100 ms/scan, collision energy 20 eV, mass scan range 25–1000 Da) were acquired for QC samples to enable metabolite identification. ESI source parameters on the Xevo GS-XS QTOF were set as the following: capillary voltage 1.8 kV, sampling cone 40 V, source temperature 50 °C, desolvation temperature 550 °C, cone gas flow 40 L/h, desolvation gas flow 900 L/h.

### 4.5. UPLC-MS Data Processing

The raw MS data (.raw) were converted to mzML files using ProteoWizard MSConvert (version 3.0.6150, www.proteowizard.sourceforge.net/). The parameters of min SNR and min peak spacing were set as 0.1 for peak picking in ProteoWizard. The files were then processed in R (version 3.4.3), using the XCMS package for feature detection, retention time correction, and alignment. The XCMS processing parameters were optimized and set as follows: mass accuracy for peak detection = 25 ppm; peak width c = (2, 30); snthresh = 6; bw = 10; mzwid = 0.015; minfrac = 0.5. The CAMERA package was used for subsequent peak annotation. The resulting data were normalized using the support vector regression algorithm in R, to remove an unwanted system error that occurred among intra- and inter-batches. Initial metabolite identification was performed using the MetDNA algorithm. Metabolites were further identified by matching retention time with an in-house metabolite standard library. In addition, metabolite identification was carried out by matching accurate mass and experimental MS/MS data against online databases (METLIN and HMDB).

### 4.6. Statistical Analysis

All statistical analyses were performed on the R platform (version 3.4.3). Orthogonal partial least-squares discriminant analysis (OPLS-DA) was used to discover global metabolic changes between colon cancer and healthy matched tissue controls, and also between right-sided colon cancer (RCC) and left-sided colon cancer (LCC), based on R package “plsdepot”. The corresponding variable importance in the projection (VIP values) was calculated in the OPLS-DA model. Meanwhile, the Wilcoxon Mann–Whitney U test, with Benjamini–Hochberg-based false discovery rates (FDR) adjust, was performed, using the R function “p.adjust” for differential analysis between colon cancer and healthy controls. For multiple groups comparisons, the nonparametric Kruskal–Wallis rank sum test was performed to determine the significance of each metabolite, and the relevant false discovery rates (FDR) based on the *p* values were estimated in the context of multiple testing. Internal cross validations of selected potential biomarkers were performed by using random forest (RF) model, partial least-squares (PLS) model, and supported vector machine (SVM) model in R. To evaluate the classification performance, the area under the receiver operating characteristic curve (AUC) value and 95% confidence interval (CI) were computed, using the pROC package on the R platform.

## Figures and Tables

**Figure 1 metabolites-10-00257-f001:**
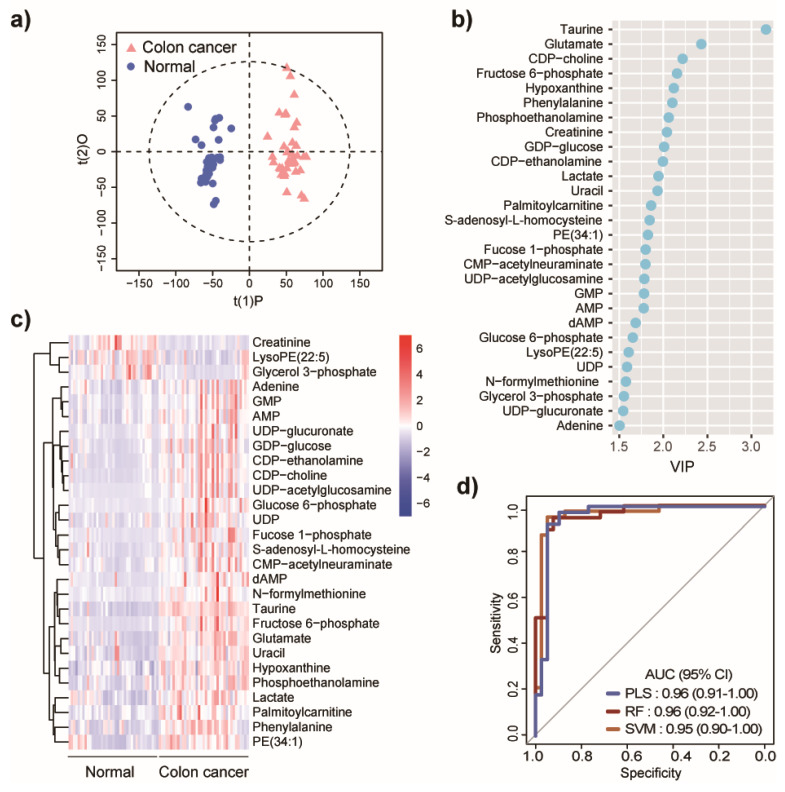
Metabolic differences in colon cancer revealed by untargeted metabolome profiling. (**a**) Orthogonal partial least-squares discriminant analysis (OPLS-DA) scores plot assessing variance of features between normal colon and colon tumors (*n* = 39/group). Dotted circle in the plot indicates the 95% confidence interval. For the OPLS-DA model, R^2^Y = 0.96, Q^2^ = 0.77. (**b**) The 28 identified metabolites from OPLS-DA analysis ranked by variable importance in the projection (VIP) values. (**c**) Heatmap showing significantly different expression levels of metabolites in cancer (vs. normal, *n* = 39) colon tissues (*n* = 39). Scale is z-score. (**d**) Receiver operating curve (ROC) plots of predictive models, based on 28 metabolic biomarkers, using partial least-squares (PLS), random forest (RF), and support vector machine (SVM), respectively.

**Figure 2 metabolites-10-00257-f002:**
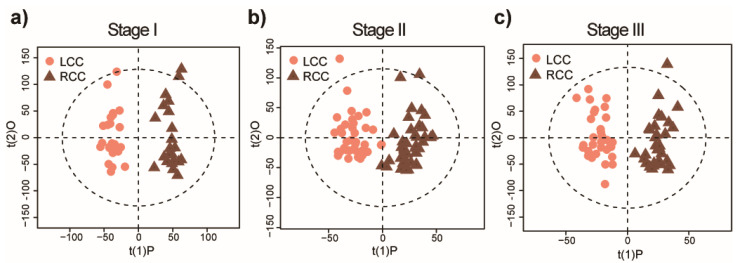
Distinct metabolomic fingerprints of left-sided colon cancer (LCC) and right-sided colon cancer (RCC). The supervised OPLS-DA plots for metabolic fingerprints of (**a**) stage I LCCs (*n* = 25) and stage I RCCs (*n* = 22) (**b**) stage II LCCs (*n* = 42) and stage II RCCs (*n* = 44), and (**c**) stage III LCCs (*n* = 32) and stage III RCCs (*n* = 32). Dotted circle in the plot indicates the 95% confidence interval.

**Figure 3 metabolites-10-00257-f003:**
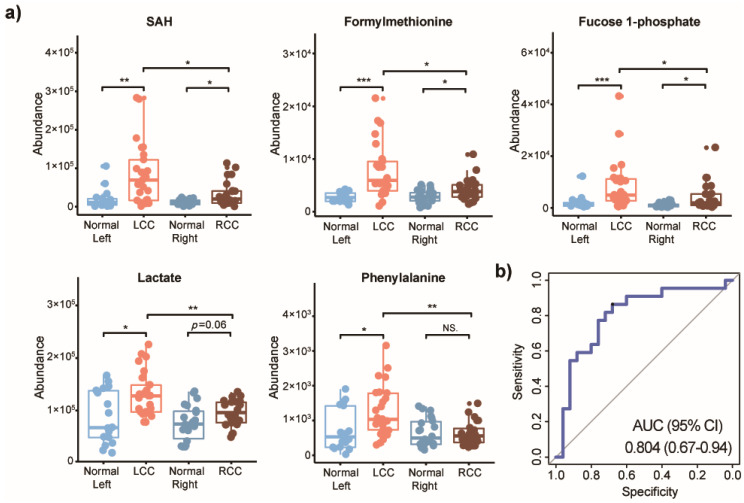
Potential metabolic biomarkers for stage I LCC and RCC. (**a**) Abundance of S-adenosylhomocysteine (SAH), formylmethionine, fucose 1-phosphate, lactate, and phenylalanine in left-sided normal colon (Normal Left, *n* = 17), left-sided colon cancer (LCC, *n* = 25), right-sided normal colon (Normal Right, *n* = 22), and right-sided colon cancer (RCC, *n* = 22). Nonparametric Kruskal–Wallis rank sum test with pairwise Wilcoxon Mann–Whitney U test, *p* values adjusted for false discovery rates (FDR) (Benjamini–Hochberg). * *p* < 0.05, ** *p* < 0.01, *** *p* < 0.001, NS. = not significant. (**b**) ROC curve plot of predictive model, based on five metabolic biomarkers using random forest (RF), differentiating LCC from RCC.

**Figure 4 metabolites-10-00257-f004:**
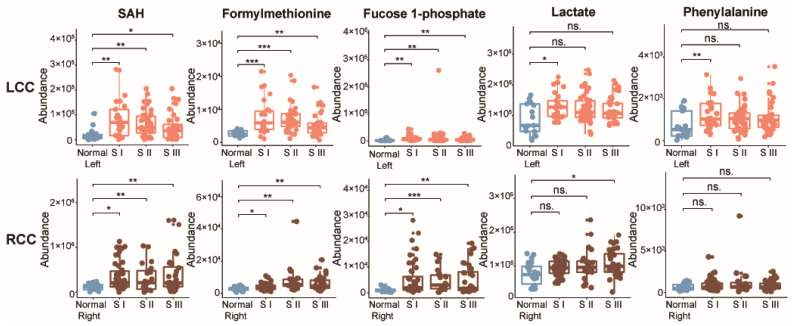
Levels of potential metabolic biomarkers in normal colon and in LCC and RCC across stages I–III. Abundance of S-adenosyl-L-homocysteine (SAH), formylmethionine, fucose 1-phosphate, lactate, and phenylalanine in left-sided normal colon (Normal Left, *n* = 17), left-sided colon cancer across stages I–III (LCC, S I: *n* = 25, S II: *n* = 42, S III: *n* = 32), right-sided normal colon (Normal Right, *n* = 22), and right-sided colon cancer across stages I–III (RCC, S I: *n* = 25, S II: *n* = 44, S III: *n* = 32). S = stage. Nonparametric Kruskal–Wallis rank sum test with pairwise Wilcoxon Mann–Whitney U test, *p* values adjusted for false discovery rates (FDR). * *p* < 0.05, ** *p* < 0.01, *** *p* < 0.001, NS. = not significant.

**Table 1 metabolites-10-00257-t001:** Potential metabolic biomarkers discriminate LCCs from RCCs at stage I.

Metabolite	m/z	RT	FC ^a^	VIP ^b^	*p* ^c^	*p* ^d^	*p* ^e^	*p* ^f^
SAH	385.1282	363.2	0.29	1.8	0.028	0.001	0.026	N.S.
Formylmethionine	176.0393	188.9	0.64	1.9	0.010	<0.001	0.033	N.S.
Fucose 1-phosphate	243.0285	438.0	0.38	1.3	0.026	0.001	0.010	N.S.
Lactate	89.0233	218.2	0.75	2.1	0.008	0.046	N.S.	N.S.
Phenylalanine	207.1130	252.0	0.53	2.3	0.003	0.019	N.S.	N.S.

^a^ FC: median value in RCCs divided by median value in LCCs. ^b^ VIP: variable importance in the projection from OPLS-DA models. ^c^
*p*: Wilcoxon Mann–Whitney U test, LCCs vs. RCCs, FDR-corrected. ^d^
*p*: Wilcoxon Mann–Whitney U test, left-sided normal colon vs. LCCs, FDR-corrected. ^e^
*p*: Wilcoxon Mann–Whitney U test, right-sided normal colon vs. RCCs, FDR-corrected. ^f^
*p*: Wilcoxon Mann–Whitney U test, left-sided normal colon vs. right-sided normal colon, FDR-corrected.

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
