# Peer review of "Tumor Tissue-Specific Biomarkers of Colorectal Cancer by Anatomic Location and Stage"

_metabolites, 2020, doi:10.3390/metabo10060257_

Round 1
Reviewer 1 Report
The paper is based on the search of tumor tissue-specific biomarkers of colorectal cancer by anatomic location and stage. This is an innovative and desirable approach to modern CRC diagnostics, prognosis and treatment because altered metabolism is a significant feature of CRC. The paper introduces the reader to the issues of the article in a clear and comprehensive way. The undoubted advantage of this study is the large number of samples analyzed. Interestingly, the authors identified 9 most discriminatory metabolites that could be used as a biomarker panel for testing in blood or urine from CRC patients. Moreover, the authors identified five metabolites (SAH, formylmethionine, fucose 1-phosphate, lactate, and 290 phenylalanine) which proved to be specific for stage I of left-sided colon cancers and this combination showed a good predictive power for differentiating between stage I RCCs and LCCs. Due to fact that RCC have worse prognosis than those with LCC clinical applicability of the five metabolites panel may prove to be useful in clinical practice and survival prognosis. Statistical analysis is carried out correctly. The discussion is short but correct. The conclusions correspond to the results.
Author Response
We very much appreciate the positive comments from the reviewer.
Reviewer 2 Report
The authors performed comprehensive untargeted metabolomics analyses on normal colon (n=39) and CRC (n=39) tissues, identifying 28 metabolite biomarkers of CRC. A stepwise selection procedure allowed to limit this signature to nine metabolites, that were the most predictive of CRC.
With a higher number of samples (n=236) they further identified five biomarkers, whose combination demonstrated high discriminating capability for left- and right-sided colon cancers at stage I.
This is an interesting study with both new and confirmatory data, the authors performed a good technical and statistical job. Yet, I noticed a potential bias and the lack of necessary information that should be added to the manuscript, to enforce its scientific value.
Major tasks:
-
Possible bias: according to methods tissue samples were collected between 1991 and 2001 and kept at -80, should I imagine that the authors have analyzed 19-29 years-old samples? Is there any published evidence that tissue samples can be stored for such long periods without metabolite decay? Did the authors perform quality controls comparing these old samples to newly sampled ones?
-
I'm not able to figure out if the 39 CRC used for the first analysis are part of the total 197 CRC analyzed in the second part of the paper, indeed the 39 normal controls are the same.
-
The authors self-cite in reference 13 a very recent paper, reporting a similar analysis (with a partially different endpoint) on 236 CRC, are these samples the same? Is the present paper a new analysis of published data? This should be clearly stated as the study would become retrospective and not prospective. Their previous study should be reported in the discussion to highlight differences and progresses made in the present study.
Important revisions:
-
CRC metabolism is strongly influenced not only by tumor location, but even more by tumor size: large tumors frequently show hypoxic areas that could be sampled. If the authors have access to the tumor size measure of their samples, I would suggest to perform a multivariate analysis including this parameter to clean results.
-
I noticed that normal samples are not evenly distributed by sex (27 males vs 12 females), as in their previous study the authors showed that sex discriminates important metabolic differences in CRC, I'm wondering if this happens also in normal samples. Could the authors highlight in figure S1 and/or fig1a the position of normal samples from males or females using a different color? This could be done only for revision, it is not necessary to include it in the paper, if differences are not outstanding.
-
As the firs part of the study lacks a validation on a second cohort, could the author use part of the 197 samples used in the second part of the paper (with a similar stage distribution) to validate their 9 CRC-specific markers? I realize that normal mucosa control would be the same, but … better than nothing.
Minor revisions:
-
Row 176 duplication: showed showed
-
Row 181 experimentally-dervied (derived?)
-
Excessive optimism in rows 222-223 and 283-285: these markers should not be reported as ready to transfer for diagnosis. Indeed, rows 229-230 show that specificity is far from reliable (68%), moreover controls from other tumor and pathological conditions should be tested before affirming that the reported signature is CRC-specific.
-
Panel d of fig 1 is apparently not cited in the results.
Reviewer 3 Report
In this manuscript, the authors identified 9 metabolites using CRC tissue extracts and extensive statistical analyses. Based on the results and analyses, the authors claimed the potential application of these metabolites for CRC diagnosis and localization (LCC vs RCC). Overall, the quality of presentation of this manuscript is good. There are some minor weaknesses that need to be addressed:
- The authors proposed that the biomarkers could be used for diagnosis using less invasive blood samples. However, levels of these metabolites were never validated in body fluids even in a small set of CRC patients. Although the authors mentioned others' research on some of the metabolites, discussion in depth will help readers gain better prospects on these biomarkers.
- There were several apparent outliers in figure 3. What were the differences in pathological features of these specimens from others?
- Are increases in five metabolic components associated with nodes and metastasis?
Author Response
Reviewer #3: In this manuscript, the authors identified 9 metabolites using CRC tissue extracts and extensive statistical analyses. Based on the results and analyses, the authors claimed the potential application of these metabolites for CRC diagnosis and localization (LCC vs RCC). Overall, the quality of presentation of this manuscript is good. There are some minor weaknesses that need to be addressed:
- The authors proposed that the biomarkers could be used for diagnosis using less invasive blood samples. However, levels of these metabolites were never validated in body fluids even in a small set of CRC patients. Although the authors mentioned others' research on some of the metabolites, discussion in depth will help readers gain better prospects on these biomarkers.
We appreciate the reviewer for the comment. We agree with the reviewer that further validation in blood samples is critical to assess these metabolite biomarkers. Therefore, we discussed the future need to explore the clinical application of the metabolites in the discussion section, which included the investigation of limits of detection in blood or urine samples and further validation in independent cohorts. In the revised manuscript, we conducted an additional analysis using another set of validation samples from the second part of the manuscript. The results demonstrated that these nine CRC metabolic markers were readily validated in this validation data set though it’s in tissue samples. We are going to recruit an additional cohort of CRC patients with blood samples available to validate in near future.
- There were several apparent outliers in figure 3. What were the differences in pathological features of these specimens from others?
We thank the reviewer for the comment. We have investigated the pathological features (tumor size, survival time, recurrence status, and gene mutation status) of these outliers in each group (Normal_Left, LCC, Normal_Right, and RCC). However, no obvious differences in clinical features were observed among these patients from others.
- Are increases in five metabolic components associated with nodes and metastasis?
We thank the reviewer for the comment. The identified five metabolites were proved to show a good predictive power for differentiating between stage I RCCs and LCCs. Since stage I tumors haven’t positive lymph nodes and metastasis, the investigation of association between these five metabolites and nodes and metastasis is not feasible.
Reviewer 4 Report
The manuscript by Cai Y et al. is interesting as there is a real and urgent need of diagnostic and prognostic biomarkers in the cancer field. However, the work has strong limitations which I detailed bellow. I also contribute with some suggestions that might be useful to the authors for the improvement of the manuscript.
Major comments
- In my opinion, in the present form the authors do not answer the main objectives set by the work: identify diagnostic and prognostic biomarkers specific for colorectal cancer in tissue specimens, to be tested/validated in body fluids in further studies.
To be truly diagnostic, the authors should have included as controls other non-tumoral alterations of the intestine such as colitis, Crohn's disease, diverticulitis, etc. The biomarker signature should be unequivocal of colorectal tumorigenesis and not shared by other intestinal conditions. Moreover no comparisons with other tumor types have been established, thus, the specificity for colorectal cancer is currently unknown.
- In the material and methods section (sample collection), the authors claim that tumor species from stage IV patients were not included in the study as their metabolism may be affected by the presence of distant metastasis, but normal adjacent tissue from these patients was used. This is counterintuitive. Also, it is not clear to me if all normal samples were derived from stage IV patients. Authors mention that normal tissue from stage I-III patients was not different from stage IV, but this is not sustained with data or previous publications. This should be clarified, as this is the basis for establishing comparisons.
- A major drawback of the work is lack of validation of the biomarker signature, both in silico and experimentally. In the first part, the authors aim to identify a set of diagnostic biomarkers discriminating between normal and tumor samples. They use 39 normal and 39 tumor samples in the discovery set. They find 28 metabolites differentially expressed, that shrink to 9 to be clinically approachable. But none of these analytes were tested in a validation set of samples, which were clearly available, as in the second part of the work the authors use a cohort of 197 tumor samples. Why not using the full amount of patients samples and split them in a discovery and validation set to assess both questions, and replicate the resuts obtained in the training set?
Regarding the experimental validation: The authors use an unbiased high though-put method such as LC-MS to identify metabolites differentially found in tumor samples. But this technology is not available at the diagnostic units. Thus, a challenge in the biomarker field is the validation of candidates using clinically affordable methods, in the case of metabolites: HPLC or biochemical determinations. Unfortunately this was not approached in the manuscript.
Minor comments
- Tables S1 and S3: provide also % as this helps the reader to determine the balance between comparison groups.
- It should be noted in the main text that the normal tissue is adjacent normal tissue from cancer patients. In the present form of the manuscript this information is only provided in material and methods section.
- QC samples should be explained/defined.
- In the first part of the study, the authors decided to perform a 7-fold cross validation instead of a “leave-one-out” or any other method. It would be nice to know the rational in the selection of bioinformatics methods used.
Author Response
Reviewer #4: The manuscript by Cai Y et al. is interesting as there is a real and urgent need of diagnostic and prognostic biomarkers in the cancer field. However, the work has strong limitations which I detailed bellow. I also contribute with some suggestions that might be useful to the authors for the improvement of the manuscript.
We thank the reviewer for their optimism for this area of study, and appreciate their suggestions.
Major comments
- In my opinion, in the present form the authors do not answer the main objectives set by the work: identify diagnostic and prognostic biomarkers specific for colorectal cancer in tissue specimens, to be tested/validated in body fluids in further studies. To be truly diagnostic, the authors should have included as controls other non-tumoral alterations of the intestine such as colitis, Crohn's disease, diverticulitis, etc. The biomarker signature should be unequivocal of colorectal tumorigenesis and not shared by other intestinal conditions. Moreover no comparisons with other tumor types have been established, thus, the specificity for colorectal cancer is currently unknown.
We appreciate the reviewer for the comment, and agree that specificity for CRC versus other intestinal diseases and cancers is extremely important. A number of studies have attempted to use untargeted metabolomics to identify body fluid metabolite biomarkers for CRC, with the ultimate aim to develop a body fluid-based diagnostic and prognostic test for screening. However, these studies have not resulted in identifying a panel of biomarkers that can be reproducibly validated between laboratories or sample cohorts. We propose that metabolite biomarker discovery would have greater success and reproducibility for CRC if the discovery phase was conducted in tumor tissues and then validated in blood samples. This would thereby eliminate any non-tumor and/or body response effects to the disease. We agree with the reviewer that further validation in blood samples is critical to assess these metabolite biomarkers. Therefore, we discussed the future need to explore the clinical application of the metabolites in the discussion section, which included the investigation of limits of detection in blood or urine samples and further validation in independent cohorts. We are going to recruit an additional cohort of CRC patients with blood samples available to validate in near future.
We do agree with the reviewer that the ideal experiment design should include controls from non-tumoral alterations of the intestine, and also comparisons with other tumor types is useful to demonstrate the specificity of biomarkers for colorectal cancer. However, due to the limited availability of patient cohorts, recent and current researches were not able to address these questions. Since metabolites are the end products and the readouts of signal transduction, it is possible to have common metabolites dysregulated in different biological contexts. For example, creatinine and glutamate which were identified in our study, were also indicated to be increased in Ulcerative colitis (UC) and Crohn’s disease (CD) (PMID:31901795). And glutamate and phenylalanine have been shown to be associated with other cancer types such as lung cancer and breast cancer (PMID:29383200, 32201524). In this study, we would like to point out the panel of nine metabolites is predictive for CRC and the panel of five metabolites when used together are able to differentiate RCC and LCC. We have taken the reviewer’s comment into consideration and have this scientific question discussed in the revised manuscript.
- In the material and methods section (sample collection), the authors claim that tumor species from stage IV patients were not included in the study as their metabolism may be affected by the presence of distant metastasis, but normal adjacent tissue from these patients was used. This is counterintuitive. Also, it is not clear to me if all normal samples were derived from stage IV patients. Authors mention that normal tissue from stage I-III patients was not different from stage IV, but this is not sustained with data or previous publications. This should be clarified, as this is the basis for establishing comparisons.
We thank the reviewer for the comment. The stage IV normal tissue samples were not excluded to improve the sample size for statistical power. Normal adjacent tissues were from stage I-IV patients. The metabolic profiles of normal tissues taken from stage IV patients were similar to the normal colon tissues taken from stage I-III patients. This can be seen in the principal components analysis (PCA) plot, which is now added to the Supplementary Information as Figure S8.
- A major drawback of the work is lack of validation of the biomarker signature, both in silico and experimentally. In the first part, the authors aim to identify a set of diagnostic biomarkers discriminating between normal and tumor samples. They use 39 normal and 39 tumor samples in the discovery set. They find 28 metabolites differentially expressed, that shrink to 9 to be clinically approachable. But none of these analytes were tested in a validation set of samples, which were clearly available, as in the second part of the work the authors use a cohort of 197 tumor samples. Why not using the full amount of patients samples and split them in a discovery and validation set to assess both questions, and replicate the results obtained in the training set?
We appreciate the reviewer for the comment. We have followed the reviewer’s suggestion on using another set of validation samples from the second part of the manuscript. The number and the sex and stage distribution of this validation patients are the same as in discovery patients. The results demonstrated that these nine CRC metabolic markers were readily validated in this validation data set. We have added the results in the revised manuscript.
- Regarding the experimental validation: The authors use an unbiased high though-put method such as LC-MS to identify metabolites differentially found in tumor samples. But this technology is not available at the diagnostic units. Thus, a challenge in the biomarker field is the validation of candidates using clinically affordable methods, in the case of metabolites: HPLC or biochemical determinations. Unfortunately this was not approached in the manuscript.
We thank the reviewer for the comment. The purpose of this study was to identify the biomarkers first using untargeted LC-MS based metabolomics. LC-MS is currently used in the clinic to screen for metabolite panels, a good example of this is inborn error of metabolite screening in neonates (amino acid disorders, urea cycle disorders, mitochondrial fatty acid beta-oxidation defect, organic acidemias, and so forth). Moreover, LC-MS can detect multiple disorders with common sample preparation and a single sample injection, which is challenging by using biochemical methods.
Minor comments
- Tables S1 and S3: provide also % as this helps the reader to determine the balance between comparison groups.
We thank the reviewer for the comment. The % have been added in the revised supplementary information.
- It should be noted in the main text that the normal tissue is adjacent normal tissue from cancer patients. In the present form of the manuscript this information is only provided in material and methods section.
We thank the reviewer for the comment. We have noted in the main text that the normal tissue is adjacent normal tissues in the revised manuscript.
- QC samples should be explained/defined.
We thank the reviewer for the comment. The QC samples were explained in the method section.
- In the first part of the study, the authors decided to perform a 7-fold cross validation instead of a “leave-one-out” or any other method. It would be nice to know the rational in the selection of bioinformatics methods used.
We thank the reviewer for the comment. Both 7-fold cross validation and leave-one-out methods are two commonly used methods for model validation. Leave-one-out is actually a special 7-fold cross validation in terms of sample size. We chose 7-fold cross validation method for its high computational efficiency.
Round 2
Reviewer 2 Report
Dear Authors,
thank you to have revised your study according to my suggestions. I think this new version is improved and would allow the reader to obtain a balanced vision of important data and potential limits. I hope you will validate these results in a more recent and complete colon samples collection.
Best regards.